# Subretinal Injection Techniques for Retinal Disease: A Review

**DOI:** 10.3390/jcm11164717

**Published:** 2022-08-12

**Authors:** Cristina Irigoyen, Asier Amenabar Alonso, Jorge Sanchez-Molina, María Rodríguez-Hidalgo, Araceli Lara-López, Javier Ruiz-Ederra

**Affiliations:** 1Department of Ophthalmology, Donostia University Hospital (HUD), 20014 Donostia San-Sebastián, Spain; 2Biodonostia Health Research Institute, 20014 Donostia-San Sebastián, Spain; 3Department of Ophthalmology, University of the Basque Country, 48940 Leioa, Spain; 4Miramoon Pharma S.L., Avenida de Tolosa, 72, 20018 San Sebastián, Spain

**Keywords:** retinal gene therapy, subretinal injection, subretinal injection technique, vitreoretinal surgery

## Abstract

Inherited retinal dystrophies (IRDs) affect an estimated 1 in every 2000 people, this corresponding to nearly 2 million cases worldwide. Currently, 270 genes have been associated with IRDs, most of them altering the function of photoreceptors and retinal pigment epithelium. Gene therapy has been proposed as a potential tool for improving visual function in these patients. Clinical trials in animal models and humans have been successful in various types of IRDs. Recently, voretigene neparvovec (Luxturna^®^) has been approved by the US Food and Drug Administration for the treatment of biallelic mutations in the RPE65 gene. The current state of the art in gene therapy involves the delivery of various types of viral vectors into the subretinal space to effectively transduce diseased photoreceptors and retinal pigment epithelium. For this, subretinal injection is becoming increasingly popular among researchers and clinicians. To date, several approaches for subretinal injection have been described in the scientific literature, all of them effective in accessing the subretinal space. The growth and development of gene therapy give rise to the need for a standardized procedure for subretinal injection that ensures the efficacy and safety of this new approach to drug delivery. The goal of this review is to offer an insight into the current subretinal injection techniques and understand the key factors in the success of this procedure.

## 1. Introduction

Intraocular delivery has become a classic route of drug administration for the treatment of eye diseases. In particular, intracameral antibiotics are used for the prevention of postoperative endophthalmitis after phacoemulsification [1] and intravitreal injections are the standard of care for the treatment of age-related macular degeneration (AMD) and other posterior pole diseases [2]. The eye offers to the practitioner certain advantages for the administration of drugs: (1) it is a privileged immune environment based on the absence of lymphatic vessels and the existence of a very tight blood–retinal barrier, (2) ocular structures are easily accessible to the clinician, (3) real-time monitoring of the structure and function of the eye is feasible, due to the large spectrum of diagnostic tools available, (4) low concentrations of drug are sufficient to achieve therapeutic doses, and finally, (5) the fellow eye can be used as a comparison group [3,4].

Intravitreal drugs are the first choice in current clinical practice when trying to reach the posterior pole, providing a high drug concentration in the vitreous cavity. Intravitreal delivery has shown efficacy treating exudative AMD, macular edema in the context of vein occlusion, and diabetic retinopathy, among other conditions [2]. Further, it is an easy procedure and has a good safety profile but it is associated with the development of endophthalmitis, elevated intraocular pressure (IOP), cataract formation, choroidal and vitreous hemorrhage, and retinal detachment [5]. It is also important to take into account that as a consequence of the barrier effect attributed to the internal limiting membrane (ILM) and Müller cells [6], intravitreal drugs do not achieve optimal concentrations in the outer retina and retinal pigment epithelium (RPE).

For diseases affecting the photoreceptor (PR) layer and the RPE, such as most inherited retinal dystrophies (IRDs), there is a need to identify a more efficient way of delivering drugs. Subretinal injection has been proposed as a suitable approach for treating these conditions. Accessing the subretinal space (SRS), the potential area between the neurosensory retina and the RPE, allows direct contact of the drug with the PR and RPE layers, optimizing the concentration of the drug in these cells. In comparison to the intravitreal space, the SRS is a safer area as it is anatomically closed and immune privileged. Moreover, smaller therapeutic drug doses are needed using the subretinal approach [7]. Recently, the ongoing evolving field of gene therapy for the treatment of retinal dystrophies has led to injection of adeno-associated virus (AAV) vectors into the SRS. This modality of treatment has shown efficacy in clinical trials treating RPE65-associated Leber congenital amaurosis (LCA) [8], CHM-associated choroideremia [9,10], CNGA3-related achromatopsia [11], and MERTK-associated retinitis pigmentosa [12]. Current cell therapies for dry AMD and Stargardt diseases are using this approach to reach the SRS [13].

It is essential to note that subretinal injection implies the creation of iatrogenic retinal detachment (Figure 1). Nevertheless, when performed with care, the technique is associated with minimal trauma and early retinal structure and function recovery, suggesting a good safety profile overall [14].

Several reports have been published on different subretinal injection approaches, all effective in the delivery of a drug into the SRS [15]. Efforts are now needed to standardize the procedure. Seeking to contribute to this process, the goal of this review is to examine the current subretinal injection techniques and understand the key factors in their efficacy and safety.

## 2. Search Strategy and Selection Criteria

Studies cited in this review were identified by searching the PubMed database for articles published between 1993 and 2022 using the following keywords: retinal gene therapy, subretinal injection, subretinal injection technique, vitreoretinal surgery, and combinations thereof. The results were then selected by title and abstract including only those studies of pertinence to our review (Figure 2).

## 3. Background

The first reports in the literature of subretinal injection appeared with macular translocation surgery [16,17]. This type of surgery implied the creation of iatrogenic retinal detachment to relocate the macula away from choroidal neovascularization, at a site with underlying healthy RPE and choroid [18]. Air was injected into the SRS to deliberately create iatrogenic retinal detachment.

In the early 2000s, subretinal injection of recombinant tissue plasminogen activator (rTPA) combined with gas tamponade emerged as an option to treat subretinal macular hemorrhage (SMH). A lack of clarity concerning the extent of penetration of the intravitreal rTPA through the retina led to the injection of this molecule through a 36-gauge (G) needle directly into the SRS [19]. The procedure resulted in the displacement of SMH secondary to exudative AMD, improving visual function and allowing potential subsequent treatments [20,21]. More recently, subretinal tenecteplase and aflibercept have been used in the therapy of SMH secondary to polypoidal choroidal vasculopathy [22,23].

In recent years, inhibitors of vascular endothelial growth factor (VEGF) have been applied subretinally. Yao et al., in a randomized clinical trial in patients with proliferative diabetic retinopathy, compared the intravitreal and subretinal injection of the VEGF inhibitor conbercept. All eyes were filled with silicone oil. Visual function was better at 6 months in the subretinal conbercept group [24]. Peng et al. described a novel technique of subretinal ranibizumab injection in eyes of children with advanced vasoproliferative disorders and total retinal detachment. Subretinal ranibizumab was effective in diminishing vascular activity in these patients [25]. Further, a subretinal approach was reported for the elimination of hard exudates in diabetic patients using a 38G needle [26] and aspiration of retained subretinal perfluorocarbon liquid has been achieved using 25G to 50G cannulas [27].

The advent and development of gene and cell therapies have popularized the subretinal drug delivery approach. To this date, different types of cells have been injected subretinally to treat patients affected with retinal disorders. Current published literature has shown successful transplantation of allogenic and autologous RPE, as well as human embryonic stem cell-derived retinal pigment epithelium (hESC-RPE), into the SRS of retinitis pigmentosa and AMD subjects [28,29,30]. Furthermore, gene therapy has been put forward as a promising approach for restoring visual function in patients with IRDs, dystrophies that affect nearly 2 million people worldwide [31]. As of June 2022, mutations in 280 genes, mainly affecting rods, cones, and the RPE, are identified as causative of these types of dystrophies (RetNet, Retinal Information Network; https://sph.uth.edu/retnet/ (accessed on 1 June 2022)). In this context, injection of viral vectors into the SRS allows for direct contact of the vector with the PR and RPE, allowing for a more efficient transduction. Intravitreal delivery of gene therapy is a less invasive procedure and thus safer, but vitreous viral vectors do not transduce in the outer layers of the retina [32].

Early attempts at gene therapy started in the 1990s with the subretinal injection of adenovirus-based vectors. In 1994, Li et al. demonstrated the transduction of outer retinal layers after subretinal delivery of a replication-deficient adenoviral vector in murine retina [33]. Subsequently, several viral vectors were evaluated. Lentiviral vectors injected into the SRS using a transscleral approach were efficient in transducing the GFP gene in PR cells [34]. Soon afterwards, AAV vectors established themselves as the preferred choice for subretinal gene therapy. In 2001, an AAV2 carrying a wild-type RPE65 gene was found to be effective in rehabilitating retinal function in a canine model of LCA. The vector was injected using a 30G anterior chamber cannula through a pars plana approach [35]. The success of AAV2 vector delivery of the RPE65 gene in a large animal model preceded the initiation of clinical trials in humans. A ground-breaking phase III clinical trial in 2017 demonstrated that injection of subretinal voretigene neparvovec (AAV2 hRPE65v2, hereon VN) improved visual function in individuals with biallelic RPE65 mutations [8]. Shortly thereafter, the US Food and Drug Administration (FDA) approved subretinal VN (Luxturna), the first ocular gene therapy to reach the market. Ongoing clinical trials are focused on different IRDs with encouraging results.

## 4. Indications for Subretinal Injections

### 4.1. Cell Therapy

Primary degeneration of PRs or a malfunction of the RPE lead to retinal degeneration. Some of the most frequent types of retinal degeneration, such as AMD and IRDs (which include retinitis pigmentosa and Stargardt disease), are caused by the irreversible loss of these cells. Therefore, replacement of damaged tissue using cell therapy could be an interesting option [28,36]. Furthermore, in many cases, cell therapy could be a good treatment as it can be used in both genetic and acquired diseases where the irreversible cell loss would make gene therapy ineffective [37].

It is believed that a variety of progenitor and stem cells can migrate into the retinal layers and survive there in order to restore the retinal function or stimulate cell regeneration. In particular, survival and migration have been proven with human tissue, including neural and retinal progenitor cells, forebrain progenitor cells, brain-derived precursor cells, embryonic stem cell-derived retinal progenitors, RPE stem cells, bone marrow mesenchymal stem cells, and with tissues from other species including rat mesenchymal cells or progenitor cells from the swine neural retina [15].

Cell therapy is a promising treatment with interesting results obtained in animal research. The first RPE cell therapy reported took place in the early 1980s in monkeys and was later reproduced in rats [28]. From that point on, various sources of cells for cell therapy have been proposed. In particular, many cell types have been tested to treat pathologic retinal degeneration in animal models. Three months after subretinal transplantation, pre-induced adult human peripheral blood mononuclear cells in mice with degenerating retina survived and migrated. In rats, human fetal lung fibroblasts with expression of the ciliary neurotrophic factor gene can be used to prevent PR degeneration and to avoid laser-induced choroidal neovascularization subretinal transplantation of RPE, over-expressing fibulin-5 can be used. Subretinal injection using mini pigs as a model of hESC-RPE cells has shown its safety, with no cell migration or tumors affecting ocular or systemic structures [15].

Now, human clinical trials are the next challenge. There are 11 ongoing and concluded clinical trials for retinitis pigmentosa, Stargardt disease, and dry and wet AMD using hESC-RPE and other human-induced pluripotent stem cells in the form of suspension or sheets [38]. Furthermore, hESC-RPE cells implanted in the SRS have an excellent long-term safety profile for dry AMD and Stargardt disease, according to two prospective phase I/II 4-year long studies involving 18 patients [15].

### 4.2. Gene Therapy

The evolution and growth of gene therapy has turned into an encouraging option for the treatment of IRDs [39].

Due to the pathophysiology of many retinal illnesses, subretinal gene therapy has the theoretical basis to turn out to be a successful treatment. Some IRDs such as retinitis pigmentosa are considered monogenic disorders, whereas both genetic and environmental risk factors underlie in AMD, with three types of cells involved (PRs, RPE, and choriocapillaris) [40,41]. With genome editing, it is possible to modify gene expression in these diseases. AAV, a small nonpathogenic dependoparvovirus, has been the most widely used vector for delivering gene therapies to the SRS. In fact, this vector has been used for subretinal drug delivery in different animal models, showing evidence of safe and stable genetic expression [15].

Moreover, delivering DNA to the retina using AAV has been useful in clinical trials since 2008 [15]. For example, the FDA approved the first gene therapy treatment for ocular disease in 2017 for the treatment of inherited biallelic RPE65 mutation-associated retinal dystrophy by delivering AAV2-mediated VN subretinally [42]. This therapy allows hope for the recessive forms of the disorder. At the moment, there are no FDA-approved gene or stem cell-based therapies targeting other genetic subtypes of retinal diseases [37]. Currently, there are more clinical trials ongoing for treating IRDs, such as retinitis pigmentosa including the X-linked form, choroideremia [43], Leber hereditary optic neuropathy, and Stargardt disease [44], and other disorders are being targeted, including achromatopsia and Usher syndrome [45]. To date, although clinical trials have mainly focused on gene augmentation therapies, other studies have also tried to repair the mutation at the mRNA level [43].

Apart from IRDs, subretinal gene therapy could address various pathological pathways, including those related to pathological retinal angiogenesis. Many ocular diseases, for instance AMD and diabetic retinopathy, have in common a dysregulation in ocular neovascularization, which is mainly induced through VEGF and their receptors. Gene therapy has been shown as an effective option in controlling neo-angiogenesis using viral vectors, which are effective in transducing various tissues and provide longstanding treatment options. Certain serotypes of viruses, such as AAV and lentivirus, have shown promising results controlling angiogenesis in animal studies. In clinical trials, the treatment has been considered safe, lacking severe adverse effects. Nonetheless, larger clinical trials are needed [42].

The possibility of combining gene therapy with stem cell therapy appears to be promising, as the cellular loss would make gene therapy ineffective [46].

### 4.3. Submacular Hemorrhage

Fovea-involving SMH is a vision-threatening ocular pathology. Experimental models have provided evidence of various pathological pathways concerning this condition. For example, toxic effects of blood located subretinally, with early PR degeneration and subsequent cell death, have been proven. Even more, blood clots are known to produce fibrin-mediated tractional damage in different retinal layers [47]. Exudative AMD is the most frequent cause underlying SMH, but there are others such as polypoidal choroidal vasculopathy, undifferentiated and traumatic choroidal neovascularization (CNV), proliferative diabetic retinopathy and retinal arterial macroaneurysm [47]. CNV can be considered the most common cause of SMH associated with AMD [20]. Management of SMH secondary to AMD can be challenging and the best treatment is not always clear.

For some cases, the subretinal approach could be useful, as the use of a VEGF inhibitor alone is not effective in medium- to large-sized SMH. For large SMH, pars plana vitrectomy (PPV) associated with subretinal rTPA and pneumatic displacement of dense SMH with an expansile gas could be an effective treatment option (Figure 3). In order to improve best-corrected visual acuity at 12 months and capture good quality images to plan future VEGF inhibitor therapy, different studies recommend performing the procedure in the first 2 weeks after the beginning of the symptoms [20,48,49].

Nevertheless, in these cases it has been proven that long-term vision may depend on the underlying disease. Therefore, potential risks and benefits must be carefully taken into account before opting for this treatment as some research has shown no differences between subretinal and conventional vitrectomy approaches [50].

## 5. Subretinal Injection Technique

### 5.1. Animal Models

Subretinal injection has been used as the delivery approach for viral vectors in multiple animal models [15]. To date, three different approaches for subretinal injection in animals have been described in the scientific literature:

#### 5.1.1. Transcorneal Approach

The transcorneal approach is commonly used for subretinal injection in rodent eyes. In a study using rat eyes, the authors advanced a 33G blunt needle through the rodent nasal cornea near the limbus. To avoid further damage of the lens, it was displaced medially using the needle. After reaching the SRS, the injection was performed by an assistant, resulting in the creation of visible retinal detachment. The procedure is monitored in real time under direct microscope visualization and full mydriasis is needed to avoid complications [51].

After using a similar technique of transcorneal subretinal injection, Qi et al. described structural recovery of the iatrogenic retinal detachment by day 1 or 2 after the procedure [52]. Pang et al. observed structural and functional recovery 5 weeks postoperatively in the mouse retina after in vitro AAV vector transduction [53].

The transcorneal route is associated with total retinal detachment, mild corneal and lens opacity, anterior synechiae, iris haemorrhage, and damaged outer PR segments [51]. Neonatal mice show an immature status of the ocular structures and therefore this approach is not feasible in these cases, as it turns out to be difficult to obtain adequate pupil dilation. Overall, transcorneal subretinal injection appears to be an effective and safe procedure in adult mice when performed by experienced practitioners [51].

#### 5.1.2. Posterior Transscleral Approach

The posterior transscleral approach accesses the SRS through the choroid. This route has a better safety profile as it does not access the intraocular media. A sclerotomy is created using a 22.5-degree ophthalmic blade 0.5 mm away from the optic nerve. Then, a 33 G beveled cannula is inserted through the sclerotomy, with an angulation of 5–10 degrees, to reach the SRS.

Normal structural and functional profiles are observed 4 weeks after the procedure. This approach has an elevated rate of success, low rate of exclusion from the treatment, and relatively few complications. Even more, this route does not require perforation of the retina and vitreous access, which makes it a safer approach for subretinal injection [54].

#### 5.1.3. Anterior Transscleral Approach

The most popular route for subretinal injection in animal models is the anterior transscleral approach (Figure 4). Using this route, intraocular media is accessed through a sclerotomy in the limbus or pars plana [55,56,57]. In a non-inferiority study comparing the safety of subretinal and intravitreal injection of viral vectors into cynomolgus monkeys, Ochakovski et al. employed a two-step procedure for subretinal injection using an anterior transscleral approach. First, a retinotomy is created and balanced salt solution (BSS) is injected subretinally with a 41G cannula (DORC, Zuidland, The Netherlands). Afterwards, a viral vector solution is introduced through the already existing retinotomy into the subretinal bleb [58]. The use of 25G 3-port PPV with or without ILM removal before subretinal injection in monkey retinas has also been described in the literature [59,60]. This route has been shown to be safe and effective for the delivery of viral vectors into the SRS in different animal models [61,62].

New elements are being integrated to enhance the success of subretinal injection technique in animal models. The use of smaller needles protects against reflux at the retinotomy site. Pressure-regulated microinjectors allow for more accurate and constant injection volumes and hence a more controlled way of creating the subretinal bleb. Intraoperative optical coherence tomography (OCT) is now adopted as a diagnostic tool to help confirm the subretinal location of the needle and monitor the creation of the bleb [63].

In mouse models of IRDs, subretinal injections fail in as many as 50% of the eyes treated. The development of a standardized protocol-based procedure could improve the effectiveness and the safety of subretinal injection [54].

### 5.2. Subretinal Injection via Vitrectomy

Subretinal injection is carried out under retrobulbar or general anesthesia. A 3-port PPV is made, generally using standard 23 or 25G trocar systems [57]. After performing core vitrectomy, posterior vitreous detachment is induced (if not already present) with or without acetonide triamcinolone for visualization of the posterior hyaloid [64]. After that, the peripheral vitrectomy is completed and the injection site is checked. For subretinal injection in SMH, an elevated region of the retina is recommended. To reduce the pressure needed for the needle to enter the SRS, some authors propose ILM peeling around the injection site [65]. The superior vascular arcade, which avoids blood vessels, is the optimum location for gene therapy. Preoperative planning using OCT and autofluorescence imaging have to be taken into account in order to avoid areas of retinal atrophy where the retina is less susceptible to detach (Figure 5) [66].

Then, subretinal injection is performed using a 25G cannula with a 38G or 41G tip, for example, a dual-bore 41G blunt polytetrafluoroethylene (Teflon)-tipped cannula mounted within a 23G steel shaft (DORC, Zuidland, The Netherlands) [66] or a 38G/25G Subretinal PolyTip Cannula (MedOne, Sarasota, FL, USA) [64]. A blunt or sharp tip can be used. Fan et al. used a beveled tip cannula, 45-degree angled scissors were applied for beveling the tip of the cannula [64]. The appearance of pseudo-schisis, which is described as momentary hydration of the outer plexiform layers on OCT, can be brought on by beveling the subretinal cannula. Therefore, the surgeon has discretion over whether to bevel the cannula or not [64]. (Video of surgical technique, Appendix A.)

Recently, new devices have been developed for subretinal injection. Wood et al. describe the access to the SRS without vitrectomy with a nanovitreal subretinal gateway device (Vortex Surgical, Chesterfield, MO, USA) for the management of SMH [67]. This device could also be used for the subretinal delivery of gene therapies, avoiding the risk associated with general anesthesia and vitrectomy, especially in pediatric patients.

Before entering the vitreous cavity, it is important to purge any air bubbles and confirm the permeability of the cannula. Non-valved trocars are recommended to prevent damaging the subretinal cannula. The technique can be performed in one or two steps. The one-step approach was described for the first time by Bainbridge et al. and consists in directly injecting the drug into the SRS [68]. The two-step variant used by MacLaren et al. and Fischer et al. consists of inducing a BSS subretinal bleb and secondly injecting the drug subretinally. This way, the creation of a space for the correct delivery of the gene vector reduces reflux into the vitreous cavity and the chance of losing the drug during subretinal injection. This technique is of special interest in patients with choroideremia where the retinal atrophy makes it more difficult to detach the retina [10,69]. The risks of the two-step technique include accidental opening of a second retinotomy, possible trauma at the retinotomy site, mechanical overstretching of the retina inducing a macular hole, and widening of the first retinotomy with a higher risk of vector reflux. The theorical bleb space is only filled with gene vector, keeping it within the accepted bleb height, when a single injection technique is performed. Theoretically, this might lessen both the overall requirement for blebs and the frequency of difficulties [64].

The volume injected subretinally varies depending on the retinal disease: 0.3 mL of VN being delivered into the SRS for RPE65-related IRDs [70] and 0.1 mL in cases of choroideremia [10]. Sometimes, a second or third bleb is recommended for various reasons: to treat a non-adjacent retinal area [66] observed on AF imaging or to avoid macular hole formation if the bleb visualized with intraoperative OCT is under tension.

There is controversy regarding foveal detachment with subretinal injection: some authors recommend direct subfoveal detachment with the injection and bleb, while others advise against the detachment of the foveal area due to the unforeseeable behavior of the retinal tissue in IRDs (thin overlying retina, PR degeneration due to macula-off detachment, and risk of macular hole) [64]. Sometimes, in areas of localized thinning, a heavy fluid can be applied to provide internal tamponade of neural retina and prevent retinal breaks due to the subretinal pressure, like in the technique explained by Maguire et al. to safeguard the thin fovea observed in patients affected with LCA [71].

Following vector delivery into the SRS, the vitreous cavity can be irrigated in an effort to purge any refluxed AAV particles. The tamponade differs between authors, from leaving the eye filled with fluid (for minimizing AAV reflux and the threat of cataract development) [66] or using air after a fluid–air exchange [8].

At the end of the procedure, the sclerotomies are sutured with 8-0 polyglactin [8,66], to avoid vector dissemination. To reduce the immune reaction to the vector, patients are treated with oral prednisone before and after surgery [8].

The injection can be performed by two surgeons (the assistant surgeon lowering the plunger of the syringe to inject the drug and the main surgeon placing the tip of the cannula subretinally) as explained by Maguire et al. and Russell et al. [8,71]. Nonetheless, the precise injection of small volumes of solution into the subretinal area by this method is a challenge: injection speed and pressure are variable, and while additional tubing systems may be used between the syringe and cannula to reduce movement, this increases the dead space requiring larger drug volumes. Recently, other methods have been proposed for single-surgeon foot-pedal-controlled automated injections using the viscous fluid injection mode of the vitrectomy system [72,73]. An automated injection system, the MicroDose Injection Device (MedOne, Sarasota, FL, USA), has been designed for low volume ophthalmic injections into the SRS; it is connected to a vitrectomy machine to allow actuation of the syringe stopper by the surgeon via a foot pedal (Figure 6). The infusion pressure is foot-pedal-controlled, with the highest limit set to the lowest that would create a constant flow of fluid instead of a stream of droplets (generally, 12–16 psi) [66]. The foot-pedal control helps minimize excessive retinal stretch with more controlled drug delivery.

Surgical precision can be further enhanced by intraoperative OCT (e.g., using a Zeiss Rescan 7000, Carl Zeiss Meditec AG, Jena, Germany; Haag-Streit intraoperative OCT system, Haag-Streit, Switzerland; Leica Proveo microscope with integrated EnFocus OCT, Leica microsystems, Danaher, Washington DC, USA; or the Duke swept- source microscope-integrated OCT system [74]). This type of imaging offers a method for real-time visualization of subretinal drug delivery and allows surgeons to quantify and assess the success of the procedure [74]. Direct real-time imaging of the SRS allowing the evaluation of instrument depth is still challenging even with current commercially available spectral-domain intraoperative OCT systems as a consequence of shadowing artifacts secondary to intraocular surgical instruments [74]. On the other hand, intraoperative OCT allows monitoring of retinal detachment progression, including the foveal area, during subretinal injection [66]. Further, an indication of the volume of the final bleb can be obtained by measuring the intraoperative OCT scan, taken at the end of the vector injection. This would allow for the calculation of the delivered dosage. This has implications for evaluating the outcomes of clinical trials, regarding the efficacy of gene therapy [66].

Excessively deep penetration of the needle tip, with subsequent blanching of the retina, may result in damage to the RPE, suprachoroidal administration of the medication, bleeding, and obstruction of the cannula tip. On the other hand, too-shallow penetration of the needle tip may result in intraretinal hydration and retinoschisis during administration of the drug. In addition, the degree of retinal tissue atrophy at the injection site may interfere with such complications and be associated with more instances of hemorrhage and cannula tip blockage. Numerous efforts to create a subretinal bleb may result in localized retinal neurosensory trauma and retinotomy widening. All these features are best assessed using intraoperative OCT, reinforcing the value of this tool for subretinal injection in gene therapy surgery [75].

Vasconcelos et al. analyzed intraoperative OCT images during subretinal gene therapy in 19 eyes. They conclude that this type of imaging offers valuable real-time feedback on cross-sectional structure of the retina while performing subretinal gene therapy surgery. Fleur-de-lis sign identification can confirm the bleb formation onset (Figure 7). The characteristics of the material used (such as the type of needle tip), as well as the surgeon’s experience, have an effect on the creation of the bleb and the presence of visible open retinotomy; a sharp needle requires partial retina and RPE/choroid indentation and a blunt needle complete indentation to create the subretinal bleb. A sharp needle was associated with more open retinotomies.

Visualization of a double hyperreflective sign with intraoperative OCT can also be useful to identify air bubbles. Another important feature of dynamic real-time intraoperative OCT is the ability to discriminate schisis from subretinal fluid. These examples show how intraoperative OCT may change surgeons’ performance, allowing them to have more control over the injection technique, switch to a different treatment area or continue the procedure. Future advances in intraoperative OCT technology, from better depth performance, retinal coverage, and ways to calculate bleb volume to the inclusion of other types of imaging study would help to improve and develop both the execution of gene therapy delivery techniques and the assessment of outcomes [75].

In the technique described by Xue et al., the retinotomies self-sealed postoperatively without complications and subretinal fluid resolved within 24 hours. Anatomical and functional recovery of the retina after iatrogenic detachment of the macula mainly occurred within the first 4 weeks after the procedure [66]. Simunovic et al. described recovery of structure and function after macular detachment for gene therapy in five men with a diagnosis of choroideremia; although retinal anatomy and visual acuity improved within 1 month, this was not accompanied by improvements in threshold sensitivity or color discrimination [14]. Future studies using advanced imaging devices, such as adaptive-optics scanning laser ophthalmoscopy and OCT, will help understand the causes of these discrepancies in visual function parameters [14].

### 5.3. Suprachoroidal Technique

Since it directly targets the choroid, RPE, and retina, the suprachoroidal space (SCS), a potential space between choroid and sclera, is a desirable location for intraocular drug administration, as it results in high drug bioavailability in these areas while maintaining low levels elsewhere in the eye. Up until recently, a significant obstacle inhibiting the general application of this strategy was the absence of a mechanism to access the SCS safely. Sclerotomy has typically been used in preclinical settings to gain access to the SCS, and the only FDA-cleared technique is a sclerotomy with micro-cannulation (iScience catheter, Ellex Medical, Adelaide, Australia). Full-thickness scleral incisions, often made with a scalpel, can also be used to enter the SCS. Once the sclerotomy is finished and the scleral-choroidal junction is identified, a catheter can be pushed through the SCS under a microsurgical scope. This approach could be helpful when delivering chemotherapy to the SCS in cases of ocular tumors [76].

Needles can also be used to gain access to the SCS (Figure 8). Given the steep learning curve for this procedure, it can be difficult to establish its safety. The success rate of an injection is known to rely on the following factors: infusion pressure, gauge, particle size, and needle length [76]. The evidence on IOP and injection success rate is controversial; some authors postulate that there is no correlation between IOP and injection success [76] but others have found an elevated IOP to be associated with a higher infusion success rate, though recognizing that it is not necessary for injection or desirable in patients [77]. They suggested that high IOP decreased scleral surface deflection, increasing microneedle insertion depth and reducing the amount of total scleral tissue between the needle tip and the SCS [77].

Overall, for the suprachoroidal technique to be considered an effective and safe therapy, technological advances are still needed. The introduction of microneedles, which are less than 1 mm long and may even be reaching nanodimensions, is a significant improvement in this field because they provide a less invasive and more reliable method of targeting the SCS than conventional needles. Further, a drug’s precise location in the SCS, precisely beneath the sclera and above the choroid, may avoid damage to the underlying retinal layers. For these reasons, ideally the length of the needles used for this purpose should be equivalent to the thickness of the conjunctiva and sclera [78], being physiologically incapable of performing an unintentional intravitreal injection deeper than the SCS [76].

Microneedle injection into the SCS is safe and effective, and it may be carried out in an outpatient ophthalmology clinic while the patient is under local anesthesia, according to ongoing clinical trials. By holding the microneedle perpendicular to the level of the scleral surface and setting the hard stop on the needle hub to contact the sclera/conjunctiva, the insertion depth may be precisely regulated. The microneedle may be left in place for up to one minute in order to reduce reflux from the injection site [76].

To access the SCS without vitrectomy, a microscope-integrated OCT could be used. Sastry et al. employed microscope-integrated OCT to precisely and accurately estimate the amount of subretinal blebs after administering BSS via a suprachoroidal cannula into ten porcine eyes. Eighty percent of the targeted injection volume could be delivered to the SRS using the suprachoroidal microscope-integrated OCT-guided method. Their results suggest that the use of subretinal delivery methods combined with a microscope-integrated OCT may increase the success of subretinal drug delivery [74].

Numerous phase III clinical trials are studying the use of microneedles for medication delivery. The most often investigated indication in both preclinical and clinical trials has been non-infectious uveitis. A drug-containing fluid is injected into this area, where it distributes circumferentially through the SCS and bathes the choroid [76]. AAV vectors are the most extensively studied viral vectors in the field of suprachoroidal gene therapy because of the reported safety and effectiveness in human clinical trials [78]. It was found in one study in which AAV8 was injected into the SRS or SCS of rhesus macaques using transscleral microneedles that suprachoroidal AAV8 causes RPE cells to express themselves widely and peripherally while also inducing a local presence of inflammatory cells. As this technique avoids vitreoretinal surgery and places less strain on the blood–retinal barrier, this method was linked to a lesser overall humoral immune response [79]. This has also been studied in rats [80].

## 6. Subretinal versus Intravitreal versus Suprachoroidal Delivery

Gene therapy is typically administered by intravitreal, subretinal, or suprachoroidal injection for ocular diseases, taking into account the target cell type to effectively guide the choice of delivery method. In general, subretinal injection is recommended to deliver genes to the outer retinal layers, whereas intravitreal injection is preferred to target the inner retina [81].

The vitreous and ILM act as a physical barrier to targeting the PRs and RPE, and this is an obstacle that intravitreal injections, the most frequent invasive treatment used today in ophthalmology both because of its safety and effectiveness, cannot solve [80]. Moreover, the intravitreal route may result in a more intense immune response when applying AAV vectors than the subretinal method. Additionally, when using intravitreal delivery, the vector may be diluted in the vitreous chamber, resulting in a decreased concentration in the outer retina. However, although most current vectors are injected into the SRS, there are some exceptions. For instance, in X-linked juvenile retinoschisis, injecting the vector intravitreally is the preferred method since it lowers the risk of retinal detachment and vitreous hemorrhage [39].

In subretinal injections, the injection of AAV vectors creates a high concentration of vector locally to increase the fraction of cells transducing and contributes in avoiding adaptive immune responses. This enables treatment of a specific area of the retina, such as the macula. Additionally, once the initial subretinal fluid is resolved, the hydrostatic force of a subretinal injection pumps vector suspension into the extracellular space of outer retinal layers, creating a possible reservoir of AAV particles for additional transduction [66]. On the other hand, in most cases, subretinal delivery of drugs requires a vitrectomy with its risks and complications [81]. It implies inducing a temporary iatrogenic neurosensory retinal detachment near the fovea, which must be carefully managed to avoid the emergence of a macular hole or persistent retinal detachment [39]. Other complications described include cataracts, retinal tears, and, less frequently, persistent vision loss. Inevitably, a subretinal injection separates the PR layer from the supporting RPE layer, impairing its function and survival even in healthy retinas and worsening the damage to previously injured retinas [82]. It must also be taken into account that the effectiveness of this method in treating pan-retinal illnesses is constrained by the fact that subretinally injected vectors only transduce a low proportion of outer retinal layer cells that are closely in contact with the subretinal bleb. Even more, the use of advanced surgical methods, a vitreoretinal surgeon, a specialized operating room, and a retrobulbar or general anesthesia could result in anesthetic difficulties, expensive costs, and a longer recovery [82]. Moreover, this delivery requires expertise in the technique and special training.

The third delivery route, the suprachoroidal route, has been shown to be a reliable approach in animal models and could be used when a less invasive technique is desirable [74]. It does not require retrobulbar anesthesia or PPV, it avoids complications and it provides more surface area coverage for the eye’s posterior segment. However, the SCS is not structurally a privileged immune environment and the existence of the choriocapillaris layer allows the medication to be cleared quickly, limiting the retina’s ability to be effectively stimulated by it [83].

## 7. Complications and Safety of Subretinal Injections

As noted above, many complications have been described with subretinal injections. These include maculopathy, macular holes, retinal and/or RPE tears, choroidal neovascularization, raised intraocular pressure, irreversible loss of vision, subretinal deposits, cataract, self-resolving subconjunctival and/or retinal hemorrhage, endophthalmitis, outer nuclear layer thinning, vector suspension reflux into the vitreous chamber with subsequent reduction of subretinal drug volumes, and possible immune responses targeting the viral capsids causing vitritis [65,66,83,84,85]. Further, incorrect depth evaluation can lead to undesired events such as choroidal rupture, iatrogenic intrusion of Bruch’s membrane, unintended injection of drug into the vitreous chamber or SCS, and retinal detachment [74].

Creating a sharp point on the cannula by cutting its tip obliquely so that it can easily penetrate retinal tissue is one technique to lower the risk of problems such as tears and bleeding linked with the application of excessive pressure when puncturing the retina [65]. Further, Okanouchi et al. recommend ILM removal for safer insertion of the cannula and observed that the pressure needed for successful injections is halved, from 12 to 6 psi, after removing the ILM, suggesting that the removal of this layer is an effective step in reducing retinal resistance to subretinal injections [65]. Unexpectedly, they observed that subretinal injections could now be administered by only touching the retinal nerve fiber layer’s surface with the cannula tip after the ILM was removed. For vitreoretinal surgeons, the ability to avoid penetrating the retina when delivering subretinal injections is particularly beneficial as it reduces the possibility of choroid and RPE damage [65].

Retinal atrophy is another potential complication. In one study, a subset of patients undergoing subretinal VN injection for the treatment of RPE65-mediated LCA developed progressive perifoveal chorioretinal atrophy after the procedure. To better understand how the risk of this event is influenced by ocular, surgical delivery, and vector-related variables, more research is required [86].

Overall, the complication rates are not easy to establish due to the small number of subjects treated so far. In the case of VN, 15- and 5-year follow-up studies to evaluate long-term efficacy and safety are ongoing in the USA (NCT03602820) and the European Union, respectively. On the other hand, ongoing work is being done to advance intraoperative OCT, 3D imaging systems with heads up display, and surgical robotics in order to enhance the subretinal injection technique [58]. For example, vitreoretinal surgery for macular hemorrhage has proven to be effective and safe with high-precision robot-assisted subretinal medication delivery of TPA [87].

## 8. Conclusions

Subretinal injections for retinal diseases are an emerging treatment option due to the growing and continuous development of gene therapy. In order to successfully target the RPE and PRs, a safe and reproducible technique is crucial for delivering gene therapy into the SRS. A successful surgery is key for the clinical effectiveness of subretinal gene therapy. This therapy poses new surgical challenges. More research and a longer follow-up of a greater number of patients are needed to assess the long-term safety and efficacy of these new treatments. It is worth mentioning that some studies published in databases other than PubMed and some ongoing studies yet to be published might have been missed. The financial support from different agencies to carry out more research on this less practiced, yet promising surgical treatment modality will be helpful in the near future.

## Figures and Tables

**Figure 1 jcm-11-04717-f001:**
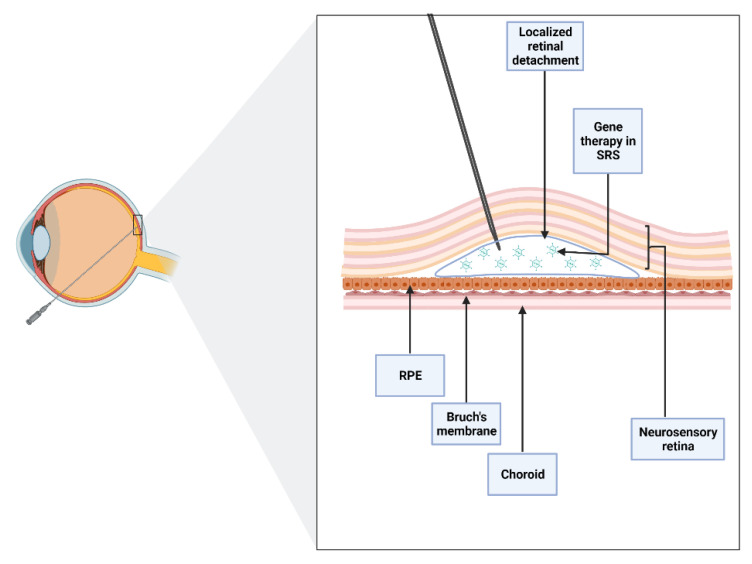
Schematic of subretinal injection. An iatrogenic retinal detachment is produced after accessing the subretinal space (SRS), the area between the neurosensory and the retinal pigment epithelium (RPE). Subretinal injection allows direct contact of the drug with photoreceptors and RPE layers.

**Figure 2 jcm-11-04717-f002:**
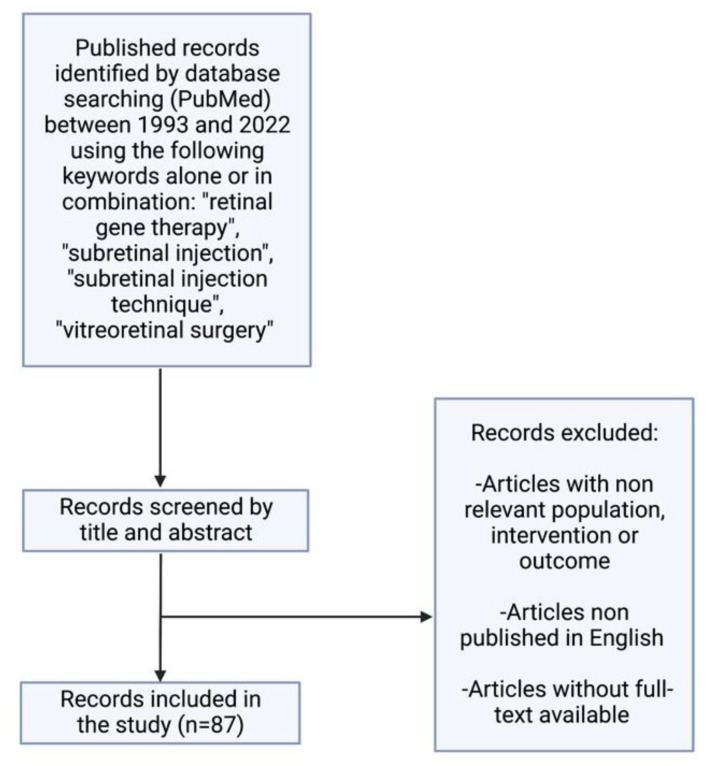
Flowchart diagram related to searching process.

**Figure 3 jcm-11-04717-f003:**
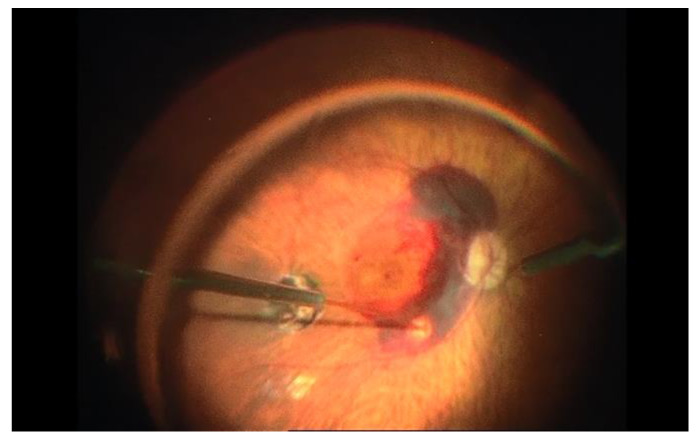
Subretinal injection of rTPA after pars plana vitrectomy in a case of large submacular hemorrhage associated with age-related macular degeneration. Subretinal injection is performed after internal limiting membrane (ILM) peeling. Note the blanching of the retina when reaching the subretinal space.

**Figure 4 jcm-11-04717-f004:**
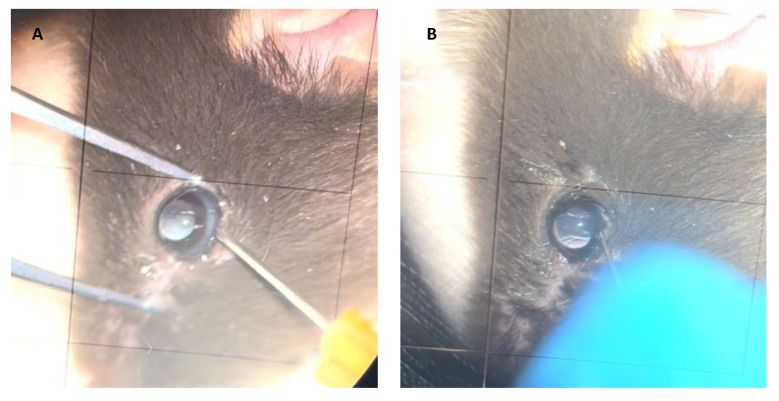
Subretinal injection technique in animal models. Anterior transscleral approach. (**A**) A retinotomy is performed in the limbus. (**B**) A blunted cannula used for subretinal injection.

**Figure 5 jcm-11-04717-f005:**
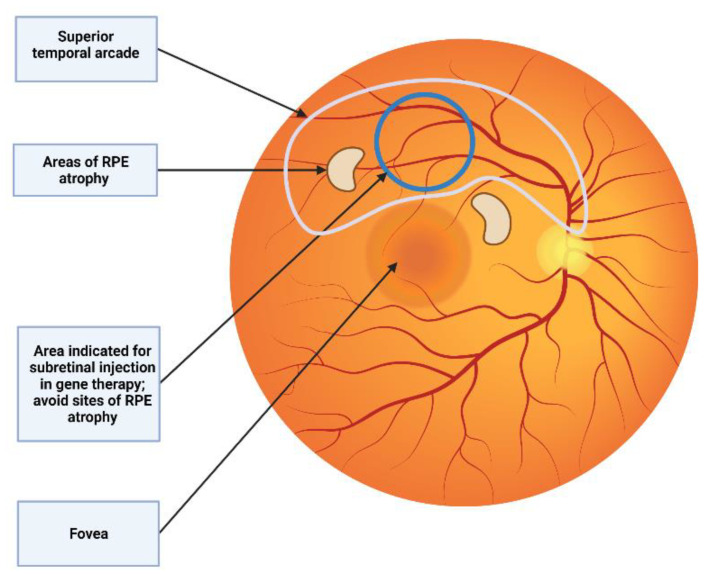
Subretinal injection location. The superior vascular arcade is the preferred location for subretinal injection. It is recommended to avoid areas of RPE atrophy where the retina is more predisposed to detach.

**Figure 6 jcm-11-04717-f006:**
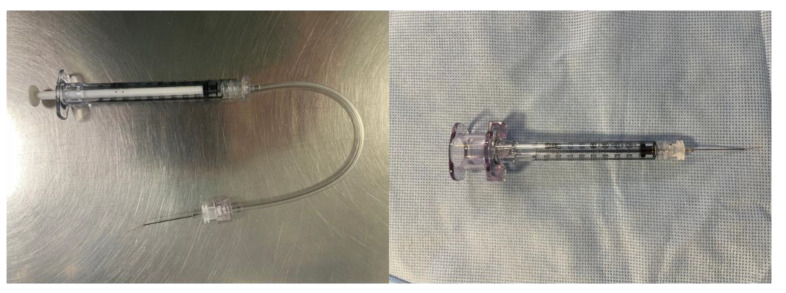
Left image shows the injection syringe and the high pressure extension tube when the injection is performed by two surgeons; when purging the treatment fluid there is loss of treatment in the extension tube. Right image shows the single-surgeon foot-pedal-controlled automated injection system using the viscous fluid injection mode of the vitrectomy system. MicroDose Injection Device (MedOne, Sarasota, FL, USA).

**Figure 7 jcm-11-04717-f007:**
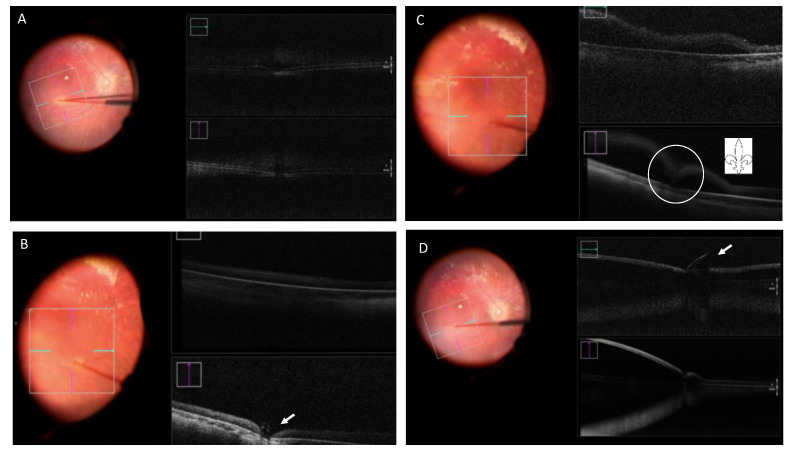
Retinography and intraoperative OCT (iOCT) taken during surgery for Luxturna gene therapy. (**A**) Blanching of the retina and indentation in the iOCT. (**B**) Indentation in the iOCT just before pressing the plunger for the subretinal injection (arrow). (**C**) “Fleur-de-lis” sign at the beginning of the subretinal injection (circle). (**D**) Subretinal bubble, localized retinal detachment and blunt cannula at the edge of the retinotomy (arrow).

**Figure 8 jcm-11-04717-f008:**
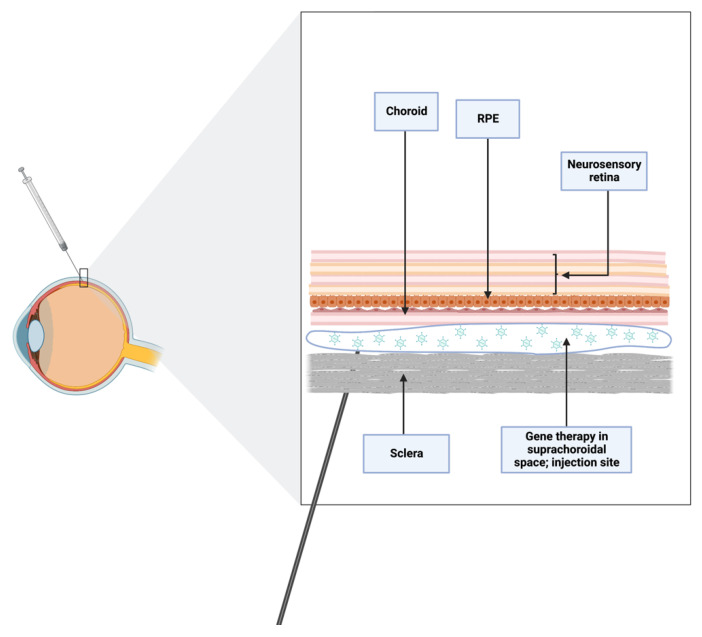
Schematic of suprachoroidal injection. Needles reaching nanodimensions are optimal for accessing suprachoroidal space, avoiding iatrogenic trauma in the vitreoretinal space.

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
