# Peer review of "Subretinal Injection Techniques for Retinal Disease: A Review"

_jcm, 2022, doi:10.3390/jcm11164717_

Round 1
Reviewer 1 Report
1. The authors mention that ‘Studies cited in this review were identified by searching the PubMed database for 82 articles published between 1993 and 2021’. It can be mentioned in the limitations that some studies published in databases other than PubMed and some ongoing studies yet to be published might have been missed.
2. The results were then screened by title and abstract to include only studies 85 that were of relevance to our review. The authors should give a flowchart mentioning this for a better understanding of the readers.
3. Since this is a review article about a challenging surgical procedure, some more pictorial illustrations like Figure 4, will add value to the manuscript.
4. The financial support from different agencies to carry out more research on this less practiced, yet promising surgical treatment modality will be helpful. This statement can be added at the end of the discussion.
Author Response
Dear Reviewer,
Thank you for your kind suggestions. We would like to answer point by point:
- We have added in the conclusions as a limitation of our study that "some studies published in databases other than PubMed and some ongoing studies yet to be published might have been missed".
- We have added a flowchart explaining the screening method of our research as "Figure 2".
- We have added another pictorial illustration as "Figure 8" illustrating suprachoroidal approach for drug/gene therapy delivery.
- We have added in the conclusions that "the financial support from different agencies to carry out more research on this less practiced, yet promising surgical treatment modality will be helpful"
Thank you
Reviewer 2 Report
This is a well written and comprehensive review of techniques to administer novel therapeutics for retinal diseases. The title could be revised as there is extensive discussion of other methods of administering these agents. The references seem adequate and the information is up-to-date.
Author Response
Dear Reviewer,
Thank you for your kind suggestions. We would like to answer to your suggestions:
We have changed the title of our review, now newly named as "Subretinal injection techniques for retinal disease: a review." as we think it better clarifies the variety of methods available for subretinal delivery of drugs to treat retinal diseases.
Thank you
This manuscript is a resubmission of an earlier submission. The following is a list of the peer review reports and author responses from that submission.
Round 1
Reviewer 1 Report
On line 347 nI would suggest that "heavy liquid " be replaced by "a small amount of perfluorocarbon liquid"
Reviewer 2 Report
Very important and relevant topic. However, there are sentences with close paraphrasing, wich should be completely avoided. Therefore I suggest to perform a major review.
During the review process, please update and correct minor errors, (e.g. the current number of gene mutations). Due to the importance of cell therapies it would be worth to include/mention earlier in the Introduction and/or in the Background section.
Please mark and annotate the Figures and their descriptions too, and clearly indicate the source too. Figures in general are not very informative, they should be replaced with more didactic images.
In the Conclusion if the method/therapy poses new challenges then standardization is not possible.